# *Lecanora* s.lat. (Ascomycota, Lecanoraceae) in Brazil: DNA Barcoding Coupled with Phenotype Characters Reveals Numerous Novel Species

**DOI:** 10.3390/jof9040415

**Published:** 2023-03-28

**Authors:** Lidiane Alves dos Santos, André Aptroot, Robert Lücking, Marcela Eugenia da Silva Cáceres

**Affiliations:** 1Programa de Pós-Graduação em Biologia de Fungos, Departamento de Micologia, Universidade Federal de Pernambuco, Campus Universitário, Recife 50670-901, PE, Brazil; ldn.stalves@gmail.com; 2Instituto de Biociências, Universidade Federal de Mato Grosso do Sul, Avenida Costa e Silva, s/n Bairro Universitário, Campo Grande 79070-900, MS, Brazil; 3Botanischer Garten, Freie Universität Berlin, Königin-Luise-Str. 6–8, 14195 Berlin, Germany; r.luecking@bo.berlin; 4Departamento de Biociências, Universidade Federal de Sergipe, Itabaiana 49500-000, SE, Brazil; mscaceres@hotmail.com

**Keywords:** South America, cryptic species, delimitation, Lecanoraceae, chemistry

## Abstract

We sequenced over 200 recent specimens of *Lecanora* s.lat. from Brazil, delimiting 28 species in our material. Many seem to represent undescribed species, some of which being morphologically and chemically similar to each other or to already described species. Here, we present a phylogenetic analysis based on ITS, including our specimens and GenBank data. We describe nine new species. The purpose of the paper is to illustrate the diversity of the genus in Brazil, not to focus on segregate genera. However, we found that all *Vainionora* species cluster together and these will be treated separately. Other *Lecanora* species with dark hypothecium clustered in several different clades. Species with the morphology of *Lecanora caesiorubella*, in which currently several subspecies with different chemistry and distribution are recognized, fall apart in different, distantly related clades, so they cannot be regarded as subspecies but should be recognized at species level. A key is given for the *Lecanora* species from Brazil.

## 1. Introduction

*Lecanora* s.lat. comprises around 1000 species of crustose lichens [1,2], with green algae as photobionts, lecanorine apothecia, and asci containing eight colourless and non-septated ascospores [3]. However, the wide variation of morphological and chemical characters found in the genus result in problematic approaches to its classification [4,5,6], and the traditional concept of *Lecanora* corresponds to different lineages [3,7,8].

The genus *Lecanora* initially comprised almost all crustose lichens with apothecia with a thalline (lecanorine) margin, regardless of the shape and colour of the ascospores. In the 20th century, the concept of the genus stabilized somewhat.

More recently, several split genera are becoming accepted within the genus, notably *Glaucomaria* M. Choisy, *Myriolecis* Clem., and *Protoparmeliopsis* M. Choisy [8], *Vainionora* Kalb [9], *Palicella* Rodr. Flakus and Printzen [10], and *Pulvinora* Davydov, Yakovch. and Printzen [11]. Except for *Vainionora*, no species are known from Brazil that belong to these split genera.

Most taxonomic work on Lecanora has been carried out in temperate or arctic environments. Only a few publications exist about *Lecanora* in the tropics [12,13,14,15,16,17], and only one of these [12] is exhaustive for a whole country.

Studies based on molecular data of problematic species and morpho-groups within *Lecanora* s.lat. have revealed several species complexes including within taxa that share phenotypic characters [3,8,18,19,20,21,22,23]. These studies have emphasized the importance of combining morphological, chemical, and molecular data in delimiting species within the genus.

In Brazil, around 60 species of *Lecanora* have been reported, not including species that belong to the segregate genus *Vainionora* (whether or not they have been combined in it). Mainly the core group has been taxonomically treated [17], although recently some additional species from partly other groups have been described [22,24,25,26,27].

In the framework of our current investigations into *Lecanora* in the wide sense in Brazil, we have already treated the representative of two genera that were formerly included in the genus, viz. *Neoprotoparmelia* Garima Singh, Lumbsch and I. Schmitt [28] and *Tephromela* M. Choisy [29]. A treatment of the representatives of the genus *Vainionora* is in preparation [30].

## 2. Materials and Methods

### 2.1. Taxon Sampling, Morphological, Terminology and Chemical Data

Approximately 450 *Lecanora s.lat.* specimens were collected during field expeditions to several areas in Brazil by the authors between 2016 and 2022. Approximately 300 were selected for molecular study.

Morphological characteristics of the apothecia and thallus were examined with an Olympus SZX7 and pictures were taken with a Nikon Coolpix 995. For analysis microscopic, hand-made sections of ascomata and thallus were mounted and studied in water. Microscopic photographs were prepared using an Olympus BX50 with Nomarski interference contrast and Nikon Coolpix 995.

Secondary metabolites were examined with thin-layer chromatography (TLC) using solvent C [31,32,33]. All the specimens were tested with UV light (λ365 nm), and spot reactions of KOH (K), paraphenylenediamine (Pd), and calcium hypochlorite (C).

The Terminology applied to distinguish Epihymenium types and Amphithecium were according to the terms explained in Guderley [17].

### 2.2. Molecular Methods

From collected specimens, approximately 300 were selected for molecular study.

Genomic DNA was extracted using the Wizard^®^ Genomic DNA Purification Kit (Promega), following the manufacturers’ protocols. For amplification of the target was used the REDExtract-N-Amp Plant PCR Sigma-Aldrich (St. Louis, Missouri, EUA). 20 µL PCR samples were prepared by adding 10 µL of PCR MIX, 4 µL of sterile water, 1 µL of each primer at 10 µM concentration, and 4 µL of DNA. PCR was performed using specifications for each marker (Table 1). PCR products were checked for amplification on 2% agarose gels. The kit Wizard^®^ SV Gel and PCR Clean-Up (Promega) was used to purify the amplified PCR products, following the manufacturer instructions. The products were then sequenced by the genetics laboratory at the Universidade Federal de Pernambuco.

### 2.3. Phylogenetic Analyses

BLAST searches (NCBI) (https://www.ncbi.nlm.nih.gov, accessed on 10 October 2022) were performed for the newly generated sequences and after confirmation of their identity, the sequences were edited and contigs were assembled using BioEdit 7.2.0 [37] and deposited in GenBank (Appendix A). For the phylogenetic analysis, we retrieved from GenBank ITS rDNA sequences classified as *Lecanora* s.lat., and then the newly generated sequences were added to the set of ITS sequences downloaded from GenBank, and aligned using MAFFT [38,39], with subsequent manual inspection. The alignment (Appendix A) had a length of 782 bases. Maximum likelihood tree search was performed using RAxML v8.1.11 [40], with GTR-Gamma and 1000 bootstrap replicates. The resulting trees were visualized in FigTree 1.4.0 [41] and subsequently edited in Microsoft PowerPoint (2016) and CorelDRAW Graphics Suite.

## 3. Results

### 3.1. Sequenced Specimens

We obtained DNA sequence data from approximately 200 specimens of the 300 which extractions were performed. Of the specimens from which we obtained sequence data, approximately 50 clustered within the *Vainionora* clade and were excluded from further analysis.

### 3.2. Phylogenetic Analyses

The dataset included 122 newly generated ITS sequences which are deposited in GenBank under accession numbers OQ650040–OQ650161 and are listed in bold, plus 61 sequences from GenBank (Appendix A). Based on morphology, chemistry, and the molecular phylogeny, our sampling included 28 species in *Lecanora s.lat*., including 10 new species (Figure 1).

In the topology found, a number of well-supported clades from sequences newly generated from known species can be distinguished within the group, and most of these clades also including new species (Figure 1).

Clade I, including *L. praeferenda* and *L. xanthoverrucosa* sp. nov.; clade II, represents *L. fluorosaxicola*, sp. nov., a single taxon on along branch with absolute support; clade III, including *L. hypocrocina*, an unidentified species (*Lecanora* sp.) from Bolívia and Brazil; *L. irregularicrocea*, sp. nov., *L. coronulans* and *L. argentata*; clade IV, including *L. pallidachroa*, sp. nov., *L. achroa, L. sulfurescens*, *L. parachroa*, *L. xanthoplumosa* and *L. notatictria*, sp. nov.; clade V, consists of four species, *L. neohelva*, sp. nov., *L. rabdotoides*, *L. plumosa* and one closely related taxon, *L.* aff. *plumosa*; clade VI, including five taxa identified as *L. brasiliana*, *L*. *saepiphila*, sp. nov., an unidentified species (*Lecanora* sp.) from this work and two lineages morphologically close to *L. polytropa* and *L. sulphurea*; clade VII, with absolute support, consists of one new taxon *L. parahelva*, sp. nov.; as well as the clade VIII, represents the new species *L. nigrilobulata*; clade IX, representing five known species, *L. subimmersa*, *L. tropica*, *L.* econorata, *L. subimmergens* and *L.* aff. *econorata*; clade X, including *L. vainioi* and *L. kalbiana*. The next five clades, consisting of pruinose species, the relationships among these clades lack support. The newly generated sequences emerge on three separate clades, one represents the new species *L. flavocaesia*, sp. nov.; and two represents *L. glaucomodes* and *L. neomerrilli*, both to be recognized at species level.

The phylogeny obtained revealed 12 new lineages corresponding to 10 new species: *L. notatictria* L.A. Santos, M. Cáceres, Aptroot and Lücking, sp. nov., *L. pallidachroa* L.A. Santos, Aptroot and M. Cáceres, sp. nov., *L. fluorosaxicola* L.A. Santos, M. Cáceres and Aptroot, sp. nov., *L. irregularicrocea* L.A. Santos, M. Cáceres, Aptroot and Lücking sp. nov., *L. flavocaesia* L.A. Santos, M. Cáceres, Aptroot and Lücking, sp. nov., *L. neohelva* L.A. Santos, M. Cáceres and Aptroot, sp. nov., *L*. *saepiphila* L.A. Santos, M. Cáceres, Aptroot and Lücking, sp. nov., *L*. *parahelva* L.A. Santos, M. Cáceres, Aptroot and Lücking, sp. nov., *L. nigrilobulata* L.A. Santos, M. Cáceres and Aptroot, sp. nov., and *L. xanthoverrucosa* L.A. Santos, M. Cáceres, Aptroot and Lücking, sp. nov. In addition, two subspecies are recognized at species level: *L. neomerrillii* L.A. Santos, Aptroot, M. Cáceres and Lücking, nom. nov. and *L. glaucomodes* Nyl.

### 3.3. Taxonomy

*Lecanora flavocaesia* L.A. Santos, M. Cáceres, Aptroot, and Lücking, sp. nov. (Figure 2).

MycoBank MB 847936.

Etymology: The epithet refers to the colour of apothecial margin.

Typus: BRAZIL, Pernambuco, Parque Nacional Vale do Catimbau, Buíque, 8°30′32.6″ S, 8°30′32.6″ S, on bark of unidentified tree, 13–15 August 2017, L.A. Santos s.n. (ISE-46521a, holotype; ITS Genbank No.OQ650143).

Diagnosis: Similar to *Lecanora caesiorubella* Ach. but differing by the distinctly yellowish, very irregular margins, and much smaller apothecia.

Description: Thallus crustose, corticolous, up to 2.5 cm wide, c. 0.1 mm thick, areolate, rimose to verruculose, whitish yellow, dull, epruinose. Soredia absent. Prothallus absent or whitish grey. Apothecia adnate, 0.8–1.5 mm diam.; disc pale orange to whitish brown, with a thin, whitish grey pruina, flat to convex; margin pale yellow, prominent, becoming excluded or persisting, strongly flexuose. Amphithecium with small crystals, small crystals rapidly dissolving in KOH (=*allophana*-type), parathecium with small crystals. Hymenium hyaline, 37.5–50 µm high; epihymenium brown to dark brown, granular with small crystals, rapidly dissolving in KOH (=*chlarotera*-type), c. 15 µm high. Hypothecium hyaline. Ascospores 8 per ascus, ellipsoid, (8–) 10–12(–13) × 5–7 μm. Pycnidia not observed.

Chemistry: Thallus and apothecia margin K+ yellow, C–, P+ orange, UV-. TLC: atranorin, protocentratic acid.

Habitat and distribution: On exposed trees in Caatinga forest; known only from Brazil.

Additional sequenced specimens examined: BRAZIL, Sergipe, Monumento Natural Grota do Angico, Poço Redondo; 9°39′39.7″ S, 37°41′09.7″ W, on bark of unidentified tree, 24 November 2016, L.A. Santos s.n. (ISE-52389)

Notes: This new species shares the chemistry of atranorin and protocetraric acid with *Lecanora caesiorubella* subsp. *glaucomodes* (Nyl.) Imshaug and Brodo but differs in the pale yellow and very irregular apothecial margins and the areolate thallus. We therefore consider the morphological differences in the apothecial margins and the thallus diagnostic.

*Lecanora fluorosaxicola* L.A. Santos, M. Cáceres, and Aptroot, sp. nov. (Figure 3).

MycoBank MB 847937.

Etymology: The epithet conveys the fluorescent yellow greenish colour of the thallus.

Typus: BRAZIL, Bahia, Estação Ecológica Raso da Catarina, Paulo Afonso, 9°39′58.8″ S, 38°28′05.2″ W, on rock, 10–12 October 2021, L.A. Santos s.n. (ISE-54033, holotype; ITS Genbank No.OQ650054).

Diagnosis: The new species can be distinguished from the other species of *Lecanora* by the fluorescent yellow greenish colour of the thallus.

Description: Thallus crustose, saxicolous, up to 7 cm wide, c. 0.1 mm thick, areolate, rimose to verruculose, yellowish green, dull, epruinose. Soredia absent. Prothallus absent or black. Apothecia adnate, 0.2–1 mm diam.; disc dark grey to greyish black, with a thick, greenish yellow pruina, flat when young to strongly concave; margin concolourous with the thallus, becoming excluded, smooth. Amphithecium with small crystals which are dissolving in KOH (=*melacarpella*-type), parathecium with small crystals. Hymenium hyaline, 42.5–62.5 µm high; epihymenium green to yellowish-green, the pigment and with small crystals soluble in KOH (= *chlarotera*-type), c. 12 µm high. Hypothecium hyaline. Ascospores 8 per ascus, ellipsoid, (9–) 10–12(–13) × 5–7.5 μm. Pycnidia not observed.

Chemistry: Thallus and apothecia margin K+ yellow, C+ yellow, P–, Thallus and apothecial disc UV+ orange. TLC: atranorin.

Habitat and distribution: On exposed rock in Caatinga forest; known only from Brazil.

Additional sequenced specimens examined: BRAZIL, Bahia, Estação Ecológica Raso da Catarina, Paulo Afonso, 9°39′58.8″ S, 38°28′05.2″ W, on rock, 10–12 October 2021, L.A. Santos s.n. (ISE-53866, ISE-53954, ISE-53921, ISE-54038).

Notes: *Lecanora fluorosaxicola* is most likely to be confused with *L. vainioi* Vänskä and *L. kalbiana* Lumbsch in the more or less brown to grey-brown apothecial disc and the substrate, as both are found on rock. The two species were described from Brazil: *L. kalbiana* was from Mato Grosso, currently only known from Brazil, and *L. vainioi* from Rio de Janeiro, restricted to the Neotropics. The new species differs in the thallus being mostly yellow, the apothecial discs with a yellowish green pruina, the hyaline hypothecium, and in the absence of zeorin. *Lecanora ryanii* T.H. Nash and Lumbsch is also superficially similar, but differs in the larger, sessile apothecia with epruinose disc; it is only known from Isla Margarita in México [42].

The differences in chemistry and morphology are reflected in the molecular phylogeny (Figure 1), where *L. fluorosaxicola* represents a single taxon on a long branch in a subclade with absolute support, whereas *L. vainioi*, *L. kalbiana* and two sequences from GenBank (Thailand) labeled *L. vainioi*, form another sub-clade. Both subclades fell in a clade with pruinose species (Figure 1).

*Lecanora glaucomodes* Nyl. (Figure 4)

Synonym: *Lecanora caesiorubella* subsp. *glaucomodes* (Nyl.) Imshaug and Brodo, Nova Hedwigia 12(1+2): 15 (1966)

Typus: Cuba, C. Wright Lich. Cub. II. 60 (H-NYL 27145, holotype)

Description specimens examined: Thallus crustose, corticolous, up to 3.5 cm wide, c. 0.1 mm thick, areolate, rimose to verruculose, whitish grey to grey, dull, epruinose. Soredia absent. Prothallus absent or whitish grey. Apothecia adnate, 0.5–1.5 mm diam.; disc pale orange to brownish orange, carneous, whitish grey pruina, flat to convex; margin slightly paler than the thallus, prominent, becoming excluded or persisting, entire to flexuose, smooth. Amphithecium with small crystals which are dissolving in KOH (= *allophana*-type), parathecium with small crystals. Hymenium hyaline, 50–75 µm high; epihymenium brown to dark brown with pigment and small crystals dissolving in KOH (= *chlarotera*-type), c. 12 µm high. Hypothecium hyaline. Ascospores 8 per ascus, ellipsoid, (10–) 10–13(–14) × 6–8 μm. Pycnidia not observed.

Chemistry: Thallus and apothecia margin K+ yellow, C–, P+ orange to red, UV–. TLC: atranorin, protocetraric acid.

Sequenced specimens examined: Brazil, Pernambuco, Parque Nacional Vale do Catimbau, Buíque, 8°31′30.6″ S, 37°14′58.8″ W, on bark of unidentified tree, 13–15 August 2017, L.A. Santos s.n. (ISE–46528a, 54328, 54329); Sergipe, Monumento Natural Grota do Angico, Poço Redondo, 9°39′39.7″ S, 37°41′09.7″ W, on bark of unidentified tree, 24 November 2016, L.A. Santos s.n. (ISE–46522, 46533, 54330, 57413); Mato Grosso, Chapada dos Guimarães, Pousada do Parque private area, 15°26′50″ S, 55°49′50″ W, “on bark of unidentified tree, 12–18 September 2020, A. Aptroot and M.F. Souza 81,989 (CGMS).

Notes: This pruinose species is chemically characterized by the presence of protocetraric acid and atranorin. Given its phylogeny, it is here recognized at species level as *Lecanora glaucomodes*. If we consider all chemical variation recently found (see Brodo et al. [43]), this material agrees partially with *L. caesiorubella* subsp. *glaucomodes*. However, the chemical definition of *L. glaucomodes* is complex. Imshaug and Brodo [44] reported atranorin and protocetraric acid as a major compound, which was found in the isotype. Later, Lumbsch et al. [45] reported that the holotype of this species produces virensic acid as a major substance. In a recent review of the *L. albella* group, it was demonstrated that the definition of species or subspecies just observing the chemistry remains complex [43]. However, the new sequences generated in the present work form a clade distinct from Genbank sequences identified as *L. caesiorubella* s.str., from the USA and Australia (Figure 1).

*Lecanora irregularicrocea* L.A. Santos, M. Cáceres, Aptroot, and Lücking sp. nov. (Figure 5).

MycoBank MB 847938.

Etymology: The name refers to the strongly flexuose apothecial margins and to differ the new species of the *coronulans* group.

Typus: BRAZIL, Rio de Janeiro, Parque Nacional de Itatiaia, Itatiaia, 22°33′ S, 42°19′ W, on bark of unidentified tree, 11–14 October 2016, M.E.S. Cáceres s.n. (ISE-38135, holotype; ITS Genbank No.OQ650060).

Diagnosis: The new species can be distinguished from the somewhat similar species

*Lecanora coronulans* Nyl. by the strongly irregularly and flexuose apothecium margin and larger ascospores.

Description: Thallus crustose, corticolous, up to 3 cm wide, c. 0.1 mm thick, areolate, rimose to verruculose, white yellowish to whitish grey, dull, epruinose. Soredia absent. Prothallus absent or whitish grey. Apothecia adnate to sessile, 0.2–1.8 mm diam.; disc brown to dark brown, epruinose, flat to convex; margin slightly paler than the thallus, prominent, flexuose, smooth, entire, sometimes strongly flexuose, whitish to whitish grey. Amphithecium with small and large crystals (= *melacarpella*-type), parathecium with small crystals. Hymenium hyaline, 50–62.5 µm high; epihymenium dark red-brown to dark brown without crystals, pigmentation not altered by KOH (= *glabrata*-type), c. 12 µm high. Hypothecium reddish brown. Ascospores 8 per ascus, broad ellipsoid, (12–) 13–17(–18) × 7–10 μm. Pycnidia not observed.

Chemistry: Thallus and apothecia margin K+ yellow, C–, P–, UV–. TLC: atranorin, zeorin.

Habitat and distribution: on bark of tree in Itatiaia rain forest; known only from Brazil.

Additional sequenced specimens examined: BRAZIL, Rio de Janeiro, Parque Nacional de Itatiaia, Itatiaia, 22°33′ S, 42°19′ W, on bark of unidentified tree, 11–14 October 2016, M.E.S. Cáceres s.n. (ISE-38120, ISE-38132).

Notes: *Lecanora irregularicrocea* represents a hitherto undescribed member of the *L. coronulans* group. All species of this group have a pigmented hypothecium and occur exclusively in the tropics [46]. This new species is similar to an assemblage of taxa that occurs in Brazil: *Lecanora concilianda* Vain., *L. concilians* Nyl., *L. coronulans* Nyl., *L. hypocrocina* Nyl. All agree in the more or less brown to dark brown apothecial disc and the *melacarpella-type* amphithecium. *Lecanora concilianda* was described from Brazil; *L. concilians* originates from Colombia, whereas *L. coronulans* and *L. hypocrocina* were described from Cuba.

The differences between the five species are mainly morphological and in part chemical: *Lecanora concilianda* and *L. concilians* differ in spore size and thallus morphology; *L. coronulans* can be distinguished by its smaller apothecia and the presence of zeorin, and *L. hypocrocina* is different from all other species by the KOH+ red to purple reaction of the hypothecium (boryquinone). The new species somewhat resembles *L. coronulans* in the presence of zeorin, but differs in the strongly irregularly and flexuose apothecium margin and larger ascospores (13–17 × 7–10 μm vs. 9.5–14 × 5–8 μm).

*Lecanora neohelva* L.A. Santos, M. Cáceres and Aptroot, sp. nov. (Figure 6)

M MycoBank MB 847939.

Etymology: The epithet is taken from the Latin neo-, meaning no, and helva, referring similar species *Lecanora helva*.

Typus: BRAZIL, Minas Gerais, Reserva Particular do Patrimônio Natural Santuário do Caraça, Catas Altas, 7°2′1″ S, 36°3′1″ W, on bark of unidentified tree, 17–25 May 2021, L.A. Santos and A. Aptroot (ISE-52342, holotype; CGMS, isotype; ITS Genbank No. OQ650118)

Diagnosis: The new species can be distinguished by the very green thallus, pale orange apothecial discs, larger apothecia and larger ascospores than in the *Lecanora helva*. The phylogenetic tree also indicate it forms an independent clade.

Description: Thallus crustose, corticolous, up to 2.5 cm wide, c. 0.1 mm thick, continuous, rimose, green to whitish green, dull, epruinose. Soredia absent. Prothallus absent or greyish black. Apothecia adnate, 0.5–1.5 mm diam.; disc pale orange to yellowish orange, epruinose, flat to convex; margin slightly paler than the thallus, prominent, becoming excluded or persisting, smooth to verruculose. Amphithecium with large and small crystals (=*melacarpella*-type), parathecium with small crystals. Hymenium hyaline, 50–75 µm high; epihymenium pale yellow to yellowish brown, with small crystals and pigment soluble in KOH (=*chlarotera*-type), c. 10 µm high. Hypothecium hyaline to pale yellowish (intensifying in KOH). Ascospores 8 per ascus, ellipsoid, (10–) 12–15 × 5–7 μm. Pycnidia not observed.

Chemistry: Thallus and apothecia margin K+ yellow, C–, P–, thallus UV+ pink, apothecia disc UV+ green. TLC: atranorin, ‘2-O-methylperlatolic.

Habitat and distribution: On exposed trees in campo rupestre; known only from Brazil.

Additional sequenced specimens examined: BRAZIL: Minas Gerais, Reserva Particular do Patrimônio Natural Santuário do Caraça, Catas Altas, 7°2′1″ S, 36°3′1″ W, on bark of unidentified tree, 17–25 May 2021, L.A. Santos and A. Aptroot (ISE-52303, ISE-52314, ISE-52319, CGMS);

Notes: This species can be recognized by the pale orange apothecial discs, the *chlarotera*-type epihymenium, and the greenish thallus. *Lecanora helva* Stizenb. is morphologically similar but can be distinguished by the smaller apothecia (0.4–1.0 mm diam. vs. 0.5–1.5 mm diam.) and smaller ascospores (5–7 × 10–13.5 μm vs. 12–15 × 5–7 μm). *Lecanora stramineoalbida* Vain. is another similar species, which differs in having a reddish brown hypothecium.

*Lecanora neomerrillii* L.A. Santos, Aptroot, M. Cáceres and Lücking, nom. nov. (Figure 7)

MycoBank MB 847940.

Synonym: *Lecanora caesiorubella* Ach. subsp. *merrillii* Imshaug and Brodo, Nova Hedwigia 12: 28. 1966. Type species—U.S.A. CALIFORNIA: Berkeley, 1893, M.A. Howe [=Cummings and Seymour, Decades of North American Lichens I, 133] (MSC, holotype).

Description specimens examined: Thallus crustose, corticolous, up to 7 cm wide, c. 0.1 mm thick, continuous, rimose to verruculose, whitish grey, dull, epruinose. Soredia absent. Prothallus absent or greyish black. Apothecia adnate, 0.5–2 mm diam.; disc pale yellow to brownish orange, carneous, whitish grey pruina, flat to convex; margin slightly paler than the thallus, prominent, becoming excluded or persisting, crenate. Amphithecium with small crystals which are dissolving in KOH (=*allophana*-type), parathecium with small crystals. Hymenium hyaline, 50–62.5 µm high; brown to dark brown with pigment and small crystals dissolving in KOH (=*chlarotera*-type), c. 12 µm high. Hypothecium hyaline. Ascospores 8 per ascus, ellipsoid, (14–) 15–18 × 8–12 μm. Pycnidia not observed.

Chemistry: Thallus K+ yellow, C–, P+ orange to red, UV–. TLC, Chemotype I: atranorin and norstictic acid. Chemotype II: atranorin and protocetraric acid.

Sequenced specimens examined: Chemotype I: BRAZIL, Minas Gerais, Reserva Particular do Patrimônio Natural Santuário do Caraça, Catas Altas, 7°2′1″ S, 36°3′1″ W, on bark of unidentified tree, 17–25 May 2021, L.A. Santos s.n. (ise–, cgms–, 52228, 52387, 52244, 52231, 52108); Chemotype II: Minas Gerais, Reserva Particular do Patrimônio Natural Santuário do Caraça, Catas Altas, 7°2′1″ S, 36°3′1″ W, on bark of unidentified tree, 17–25 May 2021, L.A. Santos s.n. (ise–, cgms– 52273); Rio de Janeiro, Parque Nacional de Itatiaia, Itatiaia, Est. Agulhas Negras, 22°22′31″ S, 44°39′44″ O, on bark of undertermined tree, 12–13 October 2016, Cáceres s.n. (ISE–38146, ISE–38149, ISE–38151).

Notes: This taxon of the *Lecanora cesiorubella* morphotype with pruinose apothecia can be separated into two chemotypes. Chemotype I, characterized by the presence of norstictic acid and atranorin; and Chemotype II, with atranorin, protocetraric acid, and undefined fatty acids, but lacking norstictic acid. Additionally, phylogenetically (Figure 1), the sequences generated for this material clustered into two distinct subclades, one with specimens that produce norstictic acid and atranorin, which agree with subsp. *merrillii* [43,44,45]. The other produces atranorin, protocetraric acid, undefined fatty acids (except the specimen ISE-52273) and lacks norstictic acid. Based on the chemistry, part of our material agrees with the *L. albella* (Chemotype I). However, *Lecanora albella* has smaller apothecia and ascospores [45], additionally, the new sequences generated in the present work form a clade distinct of Genbank sequences identified as *L. albella* (Figure 1). It is therefore recognized at species level. Therefore, we proposed the new name *Lecanora neomerrilli*, because there is already *Lecanora merrillii* Nyl, as, e.g., mentioned by Brodo et al. [43].

*Lecanora nigrilobulata* L.A. Santos, M. Cáceres and Aptroot, sp. nov. (Figure 8)

MycoBank MB 847941.

Etymology: named after the immersed, very irregular apothecia that somewhat resembles black paint spots.

Typus: BRAZIL, Minas Gerais, Reserva Particular do Patrimônio Natural Santuário do Caraça, Catas Altas, 7°2′1″ S, 36°3′1″ W, on sandstone, 17–25 May 2021, L.A. Santos and A. Aptroot (ISE-52239, holotype; CGMS, isotype; ITS Genbank No.OQ650091).

Diagnosis: *Lecanora nigrilobulata* is similar to *L. subimmersa* (Fée) Vain. and *L. oreinoides* (Körb.) Hertel and Rambold in the immersed apothecia, areolated thallus, but can be distinguished by the hypothecium colour.

Description: Thallus crustose, saxicolous, up to 3 cm wide, c. 0.1 mm thick, areolate, puzzlle shaped areoles, whitish grey, dull, epruinose. Soredia absent. Prothallus absent or black. Apothecia immersed to adnate, 0.8–1.8 mm diam.; disc black, epruinose, flat; margin concolourous with the thallus, prominent, becoming excluded or persisting, areolate. Amphithecium with large crystals which are not dissolving in KOH (= *pulicaris*-type), parathecium with small crystals. Hymenium dark brown, 50–75 µm high; epihymenium black to greenish black, the pigment turning green in KOH, without crystals (*gangaleoides*-type), c. 15 µm high. Hypothecium dark brown. Ascospores 8 per ascus, ellipsoid, (8–) 10–13(–15) × 5–7.5 μm. Pycnidia not observed.

Chemistry: Thallus and apothecia margin K+ yellow, C–, P–, UV–. TLC: atranorin.

Habitat and distribution: On exposed sandstone in campo rupestre; known only from Brazil.

Additional sequenced specimens examined: BRAZIL, Minas Gerais, Reserva Particular do Patrimônio Natural Santuário do Caraça, Catas Altas, 7°2′1″ S, 36°3′1″ W, on sandstone, 17–25 May 2021, A. Aptroot and L.A. Santos (ise–, cgms–,51230, 51622).

Notes: The new species is characterized by an areolate thallus and large, immersed apothecia with black disc and dark brown hypothecium. In the field, *L. nigrilobulata* can be confused with *L. oreinoides* (Körb.) Hertel and Rambold, but the latter differs by the shape of the areoles, which are becoming almost rounded, and the hyaline hypothecium. *Lecanora nigrilobulata* is also similar to *L. subimmersa* (Fée) Vain., but the latter has a non-areolate thallus surface, brown apothecial disc, and hyaline hypothecium.

*Lecanora notatictria* L.A. Santos, M. Cáceres, Aptroot and Lücking, sp. nov. (Figure 9)

MycoBank MB 847942.

Etymology: The name refers to the presence of notatic, isonotatic, subnotatic acids.

Typus: BRAZIL, Pernambuco, Parque Nacional Vale do Catimbau, Buíque, 8°30′20.7″ S, 37°16′34.3″ W, on bark of unidentified tree, 13–15 August 2017, L.A. Santos s.n. (ISE-46529, holotype; ITS Genbank No. OQ650041).

Diagnosis: The new species differs from the similar species *Lecanora leprosa* Fée and *L. notatica* Guderley by the chemistry and morphology.

Description: Thallus crustose, corticolous, up to 2 cm wide, c. 0.1 mm thick, areolate, rimose to verruculose, whitish grey, dull, epruinose. Soredia absent. Prothallus absent or whitish grey. Apothecia adnate, 0.3–0.8 (–1) mm diam.; disc pale yellow to brownish orange, epruinose, flat to convex; margin persistent, white, thick, entire, somewhat crenulate. Amphithecium with large crystals which are not dissolving in KOH (= *pulicaris*-type), parathecium with small crystals. Hymenium hyaline, (–37.5) 45–62 µm high; epihymenium brown to yellowish-brown with the pigment and crystals soluble in KOH (= *chlarotera*-type), c. 12 µm high. Hypothecium hyaline. Ascospores 8 per ascus, ellipsoid, (8–) 10–13 (–14) × 5–8 μm. Pycnidia not observed.

Chemistry: Thallus and apothecia margin K+ yellow, C–, PD–, UV–. TLC: atranorin, usnic acid, notatic, isonotatic, subnotatic.

Habitat and distribution: On exposed trees in Caatinga forest; known only from Brazil.

Additional sequenced specimens examined: BRAZIL, Paraíba, Reserva Particular do Patrimônio Natural Fazenda das Almas, São José dos Cordeiros, 7°28′15″ S, 36°53′51″ W, on bark of unidentified tree, 22 August 2017, L.A. Santos s.n. (ISE-46519, ISE-57412, ISE-54331, ISE-46539b, ISE-46550).

Notes: *Lecanora notatictria* can readily be identified by the pale yellow to brownish orange apothecial discs, the *pulicaris*-type amphithecium, a *charotera*-type epihymenium and the presence of atranorin, usnic acid, and notatic, isonotatic and subnotatic acids. Morphologically this species resembles *Lecanora achroa* Nyl., *L. helva*, *L. leprosa* Fée and *L. parachroa* L.A. Santos, J.G. Cavalcante and M. Cáceres, but is readily distinguished by its different chemistry. Notatic and subnotatic also occur in *L. notatica* Guderley, but this species has larger apothecia and in addition contains confluentic acid [47].

*Lecanora pallidachroa* L.A. Santos, Aptroot and M. Cáceres, sp. nov. (Figure 10)

MycoBank MB 847943.*Etymology:* The epithet is taken from the Latin *pallid*-, meaning pale, and *achroa*, referring similar species *Lecanora achroa*.

Typus: BRAZIL, Pernambuco, Parque Nacional Vale do Catimbau, Buíque; 8°30′26.7″ S, 37°18′50.9″ W, on bark of unidentified tree, 13–15 August 2017, L.A. Santos s.n. (ISE 54332a, holotype; ITS Genbank No.OQ650046).

Diagnosis: The new species can be distinguished from the somewhat similar species *Lecanora achroa* by the paler orange disc.

Description: Thallus crustose, corticolous, up to 1.5 cm wide, c. 0.1 mm thick, continuous, rimose to verruculose, whitish grey, dull, epruinose. Soredia absent. Prothallus absent or black. Apothecia adnate, 0.2–0.7 mm diam.; disc pale yellow to yellowish orange, epruinose, flat to slightly concave; margin concolourous the thallus, becoming excluded or persisting, smooth to crenate. Amphithecium with large crystals which are not dissolving in KOH (= *pulicaris*-type), parathecium with small crystals. Hymenium hyaline, 50–62.5 µm high; epihymenium brown to yellowish-brown with the pigment and crystals soluble in KOH (= *chlarotera*-type), c. 12 µm high. Hypothecium hyaline. Ascospores 8 per ascus, ellipsoid, (7.5–) 10–12 (–15) × 5–7.5 μm. Pycnidia not observed.

Chemistry: Thallus and apothecia margin K+ yellow, C–, PD–, UV–. TLC: atranorin.

Habitat and distribution: On exposed trees in Caatinga forest; known only from Brazil.

Additional sequenced specimens examined: BRAZIL. Alagoas, Pedra Talhada private area, 9°15′ S, 36°25′35″ W, ca. 500 m; on wooden pole, 21–23 October 2017, M.E.S. Cáceres and A. Aptroot (ise–, ABL–, 42602c)

*Notes: Lecanora pallidachroa* is most likely to be confused with *L. achroa*, but differs in the paler orange disc, the smaller ascospores (7.5–) 10–12 (–15) × 5–7.5 vs. 11–16 × 5–8), and the absence of usnic acid.

*Lecanora parahelva* L.A. Santos, M. Cáceres, Aptroot and Lücking, sp. nov. (Figure 11)

MycoBank MB 847944.

Etymology: The epithet refers to the superficial morphological similarity of the new species with *L. helva*.

Typus: BRAZIL, Sergipe, Monumento Natural Grota do Angico, Poço Redondo, 9°39′39.7″ S, 37°41′09.7″ W, on bark of unidentified tree, 24 November 2016, L.A. Santos s.n (ISE-54323, holotype; ITS Genbank No.OQ650113).

Diagnosis: The new species differs from the similar species *Lecanora helva* by the presence of the alectoronic acid.

Description: Thallus crustose, corticolous, up to 3 cm wide, c. 0.1 mm thick, areolate, rimose to verruculose, yellow to pale orange, dull, epruinose. Soredia absent. Prothallus absent or black to greyish black. Apothecia adnate, 0.2–1.5 mm diam.; disc pale yellow to yellowish orange, epruinose, flat to convex; margin concolourous slightly paler than the thallus, prominent, persisting, thick, entire to wavy, smooth. Amphithecium with small and large crystals (=*melacarpella*-type), parathecium with small crystals. Hymenium hyaline, 40–62.5 µm high, inspersed; epihymenium brown to yellowish-brown, the pigment and small crystals soluble in KOH (=*chlarotera*-type), c. 10 µm high. Hypothecium hyaline to pale yellowish (intensifying in KOH). Ascospores 8 per ascus, ellipsoid, (8–) 10–12 × 5–7 μm. Pycnidia not observed.

Chemistry: K+ yellow, C–, P–, medulla apothecial UV+ greenish-white, thallus UV+ orange. TLC: atranorin, 2-chloro-*6-O*-methyl-norlichexanthone, 4,5 dichloronorlichexanthone and alectoronic acid.

Habitat and distribution: On exposed trees in Caatinga forest; known only from Brazil.

Additional sequenced specimens examined: Sergipe, Monumento Natural Grota do Angico, Poço Redondo, 9°39′39.7″ S, 37°41′09.7″ W, on bark of unidentified tree, 24 November 2016, L.A. Santos s.n (ISE-57414); Pernambuco, Parque Nacional Vale do Catimbau, Buíque, 8°33′29.1″ S, 37°13′58.3″ W, on bark of unidentified tree, 13–15 August 2017, L.A. Santos s.n. (ISE-46531, ISE-46532, ISE-54324, ISE-54326, ISE-54325, ISE-46523, ISE-46530); Bahia, Estação Ecológica Raso da Catarina, Paulo Afonso, 9°39′58.8″ S, 38°28′05.2″ W, on bark of unidentified tree, 10–12 October 2021, L.A. Santos s.n. (ISE-53991, ISE-53989).

*Notes: Lecanora parahelva* is chemically characterized by the presence of alectoronic acid in addition to atranorin, 2-chloro-6-O-methyl-norlichexanthone and 4,5 dichloronorlichexanthone. Anatomically it can be recognized by the presence of yellow thallus, large apothecia with margins slightly paler than the thallus.

*Lecanora saepiphila* L.A. Santos, M. Cáceres, Aptroot and Lücking, sp. nov. (Figure 12)

MycoBank MB 847945.

Etymology: The epithet is taken from the Latin *saepi phila*, meaning fence, referring to the substrate which the new species was collected.

Typus: BRAZIL, Alagoas, Pedra Talhada private area, 9°15′ S, 36°25′35″ W, ca. 500 m; on wooden pole, 21–23 October 2017, M.E.S. Cáceres and A. Aptroot (ISE-42612b, holotype; ABL, isotype; ITS Genbank No.OQ650100).

Diagnosis: The new species differs from the similar species *Lecanora achroa* by the absence of usnic acid.

Description: Thallus crustose, corticolous, up to 1.8 cm wide, c. 0.1 mm thick, continuous, thin, white to whitish grey, dull, epruinose. Soredia absent. Prothallus absent or greyish black. Apothecia adnate, 0.3–0.8 (–1) mm diam.; disc pale orange to orange brownish yellow, epruinose, flat to convex; margin concolourous with the thallus, prominent, persisting, entire to wavy, smooth. Amphithecium with large crystals which are not dissolving in KOH (*pulicaris*-type), parathecium with small crystals. Hymenium hyaline, 42–62.5 µm high; epihymenium brown to yellowish brown, the pigment and small crystals soluble in KOH (=*chlarotera*-type), c. 12 µm high. Hypothecium hyaline. Ascospores 8 per ascus, ellipsoid, 0.8–10 × 5–6 μm. Pycnidia not observed.

Chemistry: Thallus and apothecia margin K+ yellow, C–, P–, apothecia disc UV+ green. TLC: atranorin, ‘2-O-methylperlatolic.

Habitat and distribution: On wooden pole in Pedra Talhada rain forest remant; known only from Brazil.

Additional sequenced specimens examined: BRAZIL, Alagoas, Pedra Talhada private area, 9°15′ S, 36°25′35″ W, ca. 500 m; on wooden pole, 21–23 October 2017, M.E.S. Cáceres and A. Aptroot (ise–, ABL–,42602b).

Notes: *Lecanora saepiphila* is characterized by the small apothecia with pale orange to orange brownish apothecial discs, which are UV+ green. Morphologically it is similar to *L. achroa* and *L. parachroa.* However, both species have slightly larger ascospores. Moreover, those two species can be readily distinguished from *L. saepiphila* by their different chemistry; in addition to atranorin, *L. achroa* produces usnic acid and *L. parachroa* an unidentified fatty acid [22].

*Lecanora xanthoverrucosa* L.A. Santos, M. Cáceres, Aptroot, and Lücking sp. nov. (Figure 13).

MycoBank MB 847946.

Etymology: The name refers to strongly verrucose thallus and apothecial margins and to the presence of lichexanthone.

Typus: BRAZIL, Sergipe, Monumento Natural Grota do Angico, Poço Redondo; 9°39′39.7″ S, 37°41′09.7″ W, on bark of unidentified tree, 24 November 2016, L.A. Santos s.n. (ISE-52394, holotype; ITS Genbank No. OQ650078).

Diagnosis: The new species can be distinguished from the somewhat similar species *Lecanora coronulans* Nyl. by the verrucose thallus and apothecial margins and smaller ascospores.

Description: Thallus crustose, corticolous, up to 3 cm wide, c. 0.1 mm thick, areolate to densely verrucose, whitish grey, dull, epruinose. Soredia absent. Prothallus absent or blackish grey. Apothecia adnate, 0.3–0.9 mm diam.; disc dark brown to black, epruinose, flat to convex; margin concolourous with thallus, prominent, crenate or verruculose to densely verrucose, whitish to whitish grey. Amphithecium with small and large crystals (=*melacarpella-type*), parathecium with crystals. Upper part of hymenium brownish, lower part hyaline, 50–62.5 µm high; epihymenium brownish with crystals, pigmentation not altered by KOH (=*pulicaris-type*), c. 12 µm high. Hypothecium reddish brown. Ascospores 8 per ascus, broad ellipsoid, (7.5–) 8–11(–12) × 5–7 μm. Pycnidia not observed.

Chemistry: Thallus and apothecia margin K+ yellow, C–, P–, UV+ yellow. TLC: atranorin and a xanthone, possibly lichexanthone.

Habitat and distribution: On exposed trees in Caatinga forest; known only from Brazil.

Additional sequenced specimens examined: BRAZIL, Sergipe, Monumento Natural Grota do Angico, Poço Redondo; 9°39′39.7″ S, 37°41′09.7″ W, on bark of unidentified tree, 24 November 2016, L.A. Santos s.n. (ISE-46520); Povoado Rio dos Negros, Carira; 9°29′ S, 37°48′ W, on bark of unidentified tree, 14 November 2019, L.A. Santos s.n. (ISE-52392, ISE-52393); Alagoas, Pedra Talhada private area, 9°15′ S, 36°25′35″ W, ca. 500 m; on wooden pole, 21–23 October 2017, M.E.S. Cáceres and A. Aptroot (ISE-42608c; ABL); Bahia, Lençois, Chapada Diamantina; 12˚33′33″ S, 41˚23′20″ W, on bark of unidentified tree, 21 July 2017, M.E.S. Cáceres and A. Aptroot (ise–, ABL–, 40532c)

*Notes: Lecanora xanthoverrucosa* represents another undescribed member of the *L. coronulans* group. It resembles *L. coronulans*, *L. hypocrocina* and *L. egranulosa* Nyl., but can be readily distinguished by the pigmentation of the epihymenium, the verruculose thallus, and the presence of a xanthone.

## 4. Discussion

The circumscription of species within of *Lecanora* s.lat. remains complex and many studies focusing on delimitation of groups within this genus have been inconclusive [1,8,18,21,48]. The aim of this study was to illustrate the diversity of the genus in its broad sense in Brazil, not to discuss segregation of smaller genera and the delimitation of *Lecanora* s.str. We therefore treat this group as *Lecanora* s.lat.

The few tropical *Lecanora* specimens sequenced so far are mainly from the Paleotropics [7,49] and a recent study with molecular data focusing on Bolivia [50]. With additional sequences from Brazil, we found that some of the species’ identifications of published sequences are incongruent; for example, the GenBank (GB) sequences identified as *L. vainioi* from Thailand were clustered in a distinct branch separate from the Brazilian material, the type of *L. vainioi* being is from Brazil [51]. Consequently, the Brazilian specimens are likely to represent *L. vainioi* s.str. and the two Thai GB sequences are something else. Our phylogenetic analyses shown that some morphologically similar specimens were recovered in separate strongly supported lineage and/or clades (Figure 1).

Species having a dark hypothecium did not cluster in a monophyletic group, as also demonstrated previously [7]. Nine sequenced species in our material have a dark hypothecium, and three of these are new species.

*Lecanora caesiorubella* is not monophyletic, as also demonstrated previously [7,8]. Our phylogeny shows a separation of the studied specimens into two subclades, both with two species (Figure 1), although all fit some traditionally recognized subspecies of *L. caesiorubella* [44,45]. Two of them resemble each other both chemically and morphologically, viz,. subsp. *merrillii* and subsp. *glaucomodes*. Curiously, all our sequences of pruinose specimens are positioned in a distinct branch of GB sequences from *L. caesiorubella,* viz. with *L. albella, L. farinacea,* and *L. subcarnea;* similar to Medeiros et al. [50]. Reports of *L. caesiorubella* Ach. From Brazil are probably all referable to *L. neomerrillii.*

Our data thus confirm that ITS provides good resolution for species delimitation in the genus *Lecanora.*

The genus *Lecanora* has a wide geographic distribution in Brazil (https://www.gbif.org/occurrence/map?country=BR&taxon_key=2569863, accessed on 10 December 2022), except for the North region. During a lecanoroid lichen investigation in Brazil, all regions and ecosystem of the country were visited. Of these areas, twenty-eight taxa of *Lecanora* are treated here, of which 10 are new to science. Despite being a highly fragmented ecosystem [52], most of the *Lecanora* specimens were collected in areas of the Caatinga, also including new species. Interestingly, the majority of the new species are restricted to the ecosystem areas where they were initially found. This study shows morphological and molecular data that demonstrate the species richness and provide data for a better delimitation between the species of *Lecanora* from Brazil.

## 5. Key to Species of *Lecanora* in Brazil

This key is not exhaustive; further species have been reported or described from Brazil, but their identity is not known to us, and most are probably wrong identifications. Species referable to *Vainionora* are excluded from this key; they all share a dark hypothecium and the presence of an UV+ orange to red xanthone in thallus and/or apothecium. 
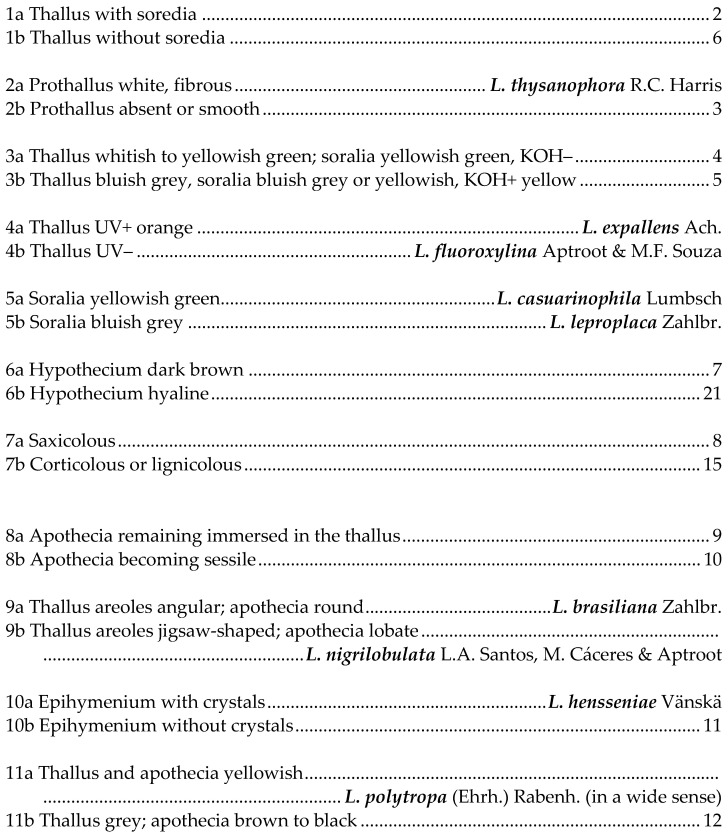


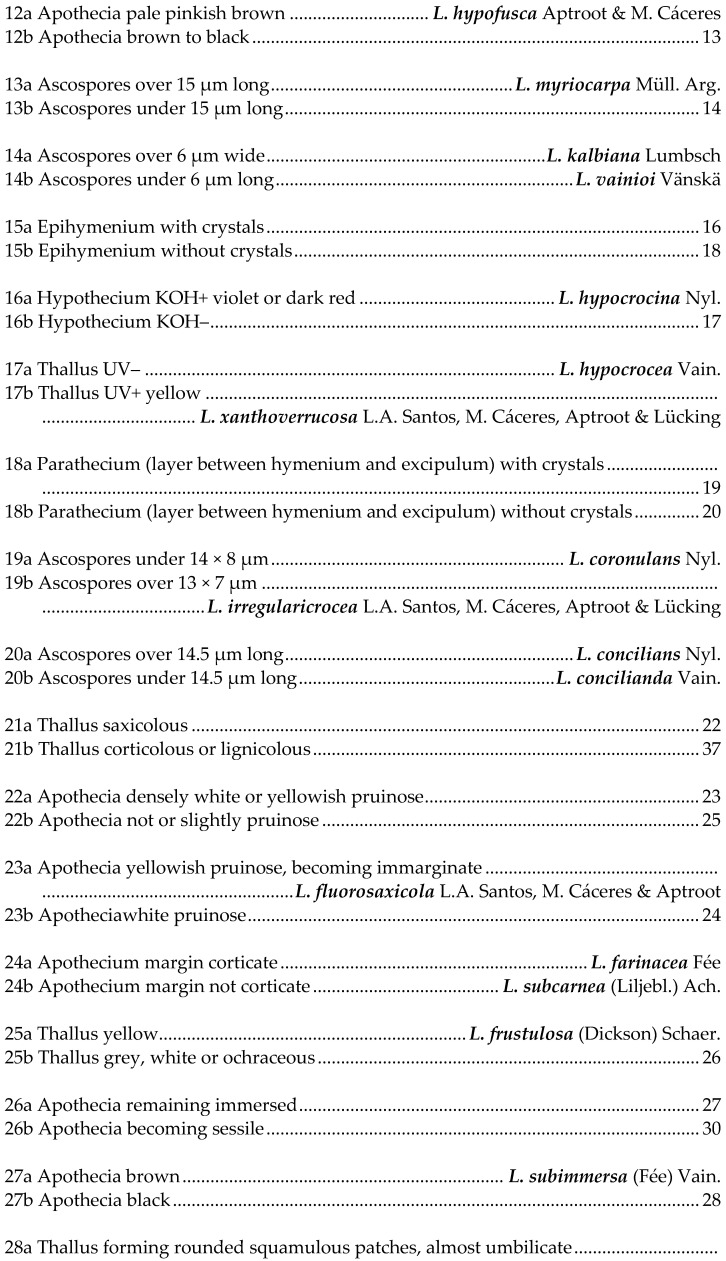


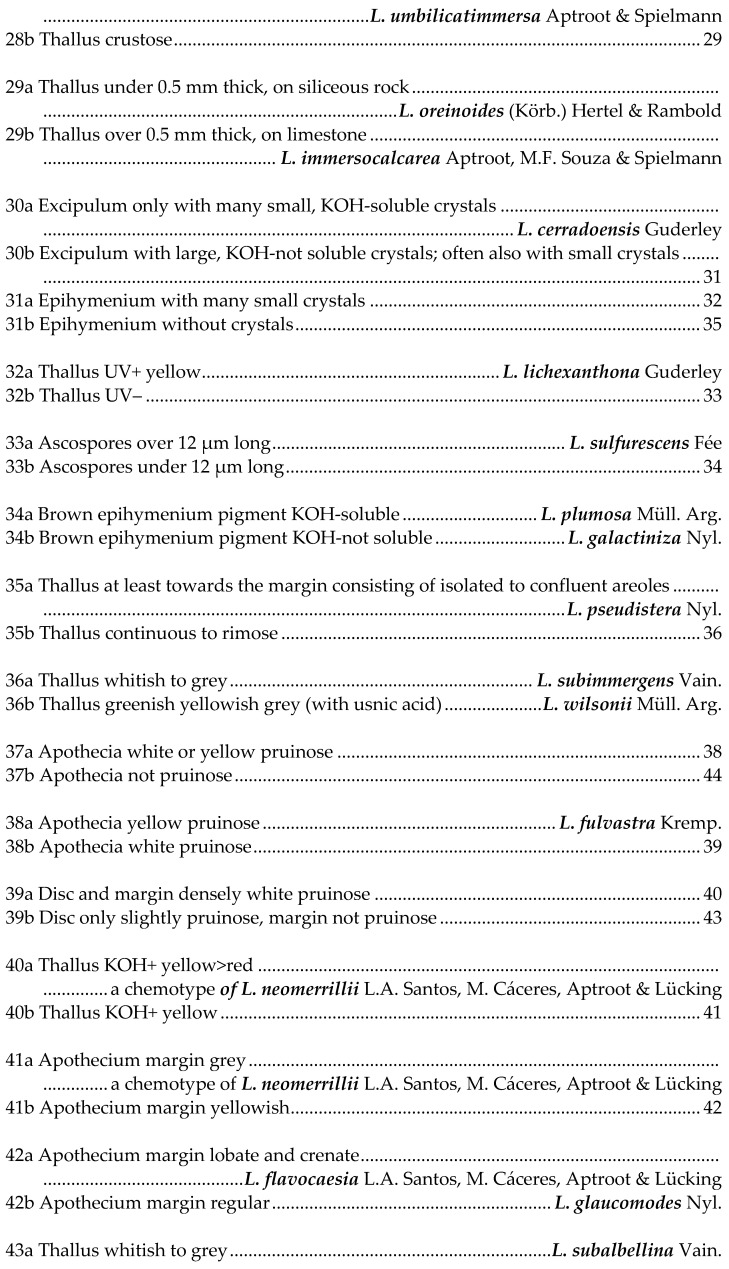


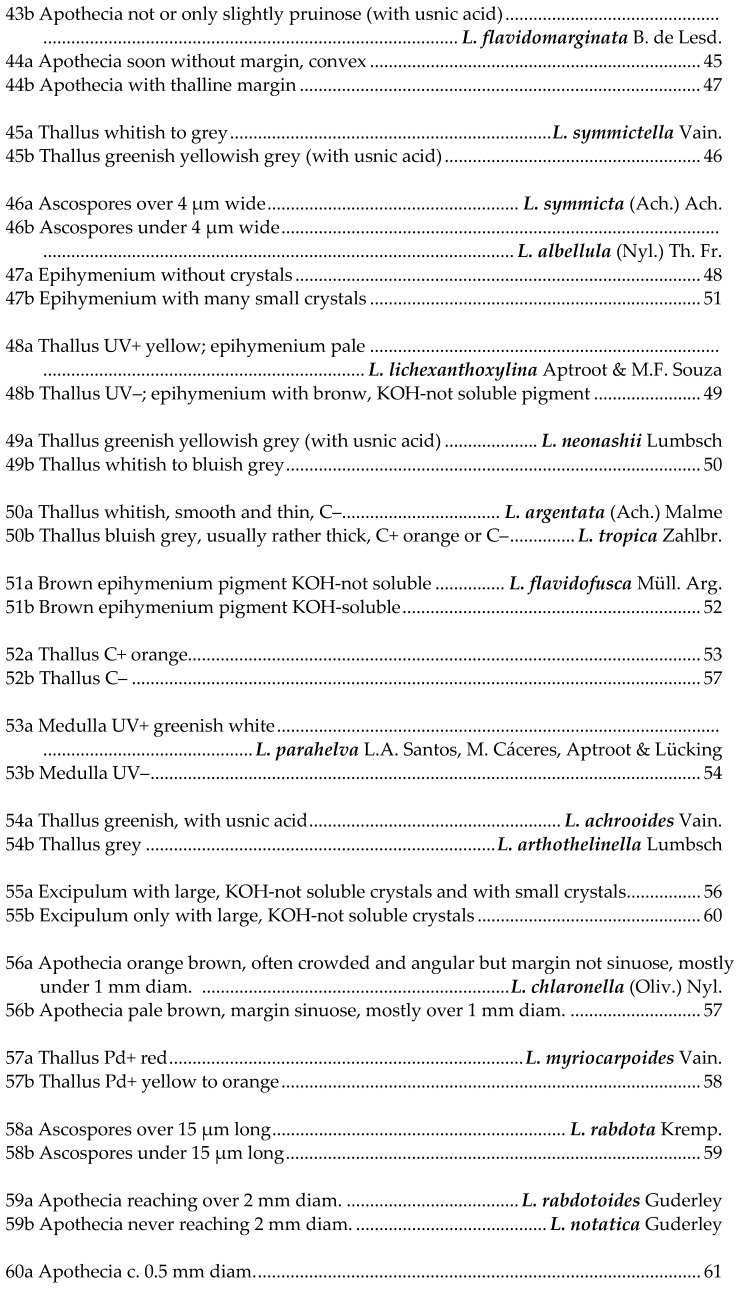


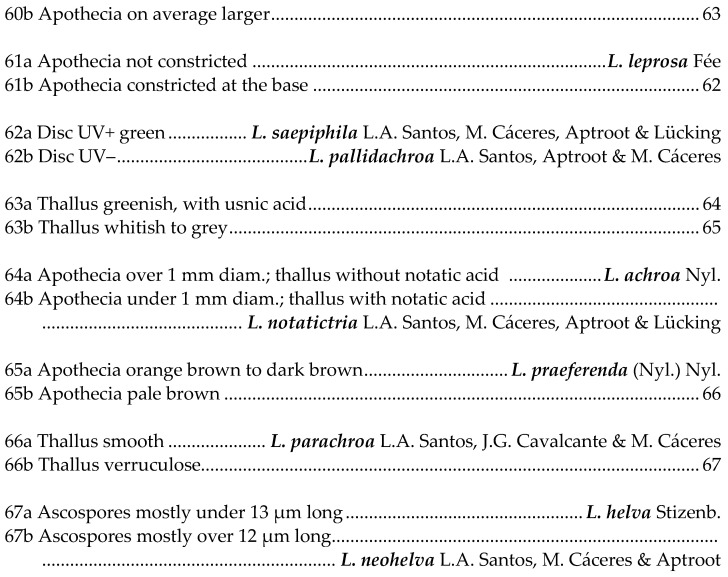


## Figures and Tables

**Figure 1 jof-09-00415-f001:**
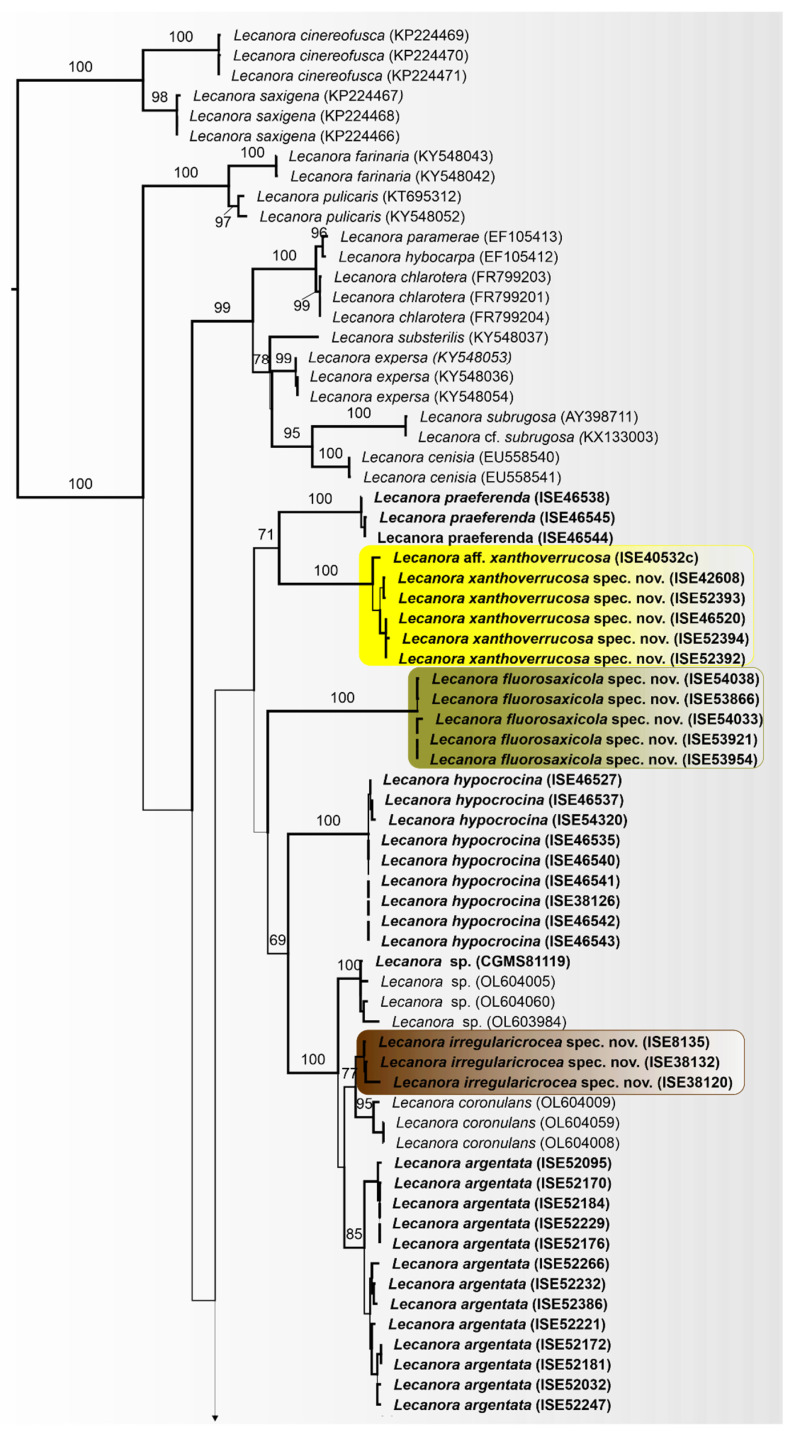
Best-scoring ML tree based on the analysis of ITS sequence data. Bootstrap support values are given above the branches. The new sequences generated in this study are indicated in bold and new *Lecanora* species is highlighted.

**Figure 2 jof-09-00415-f002:**
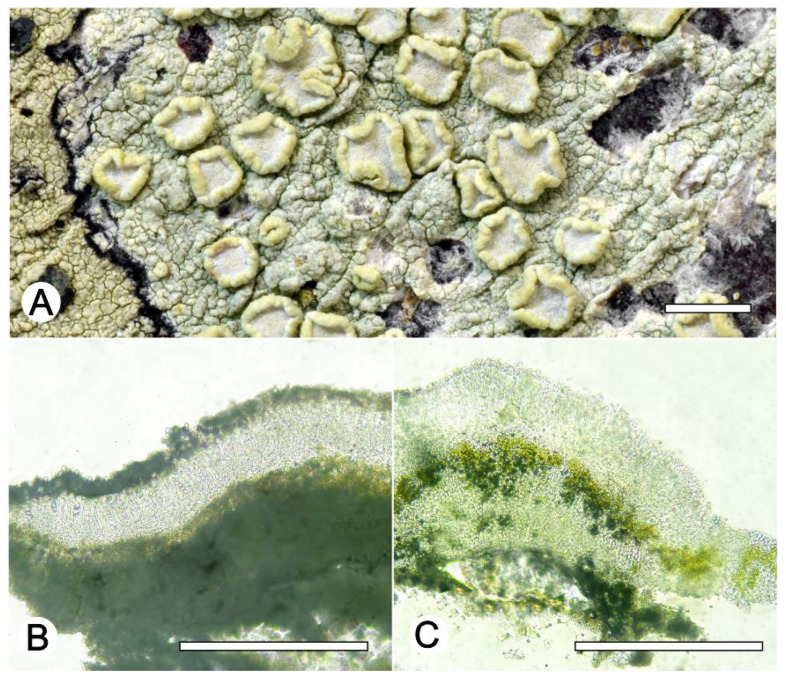
*Lecanora flavocaesia* sp. nov. (**A**) Thallus with ascomata (holotype ISE-46521a); (**B**) Apothecial section in water (holotype ISE-46521a); (**C**) Apothecial section after KOH treatment (holotype ISE-46521a). Scale bars: (**A**), 1 mm; (**B**,**C**), 100 μm.

**Figure 3 jof-09-00415-f003:**
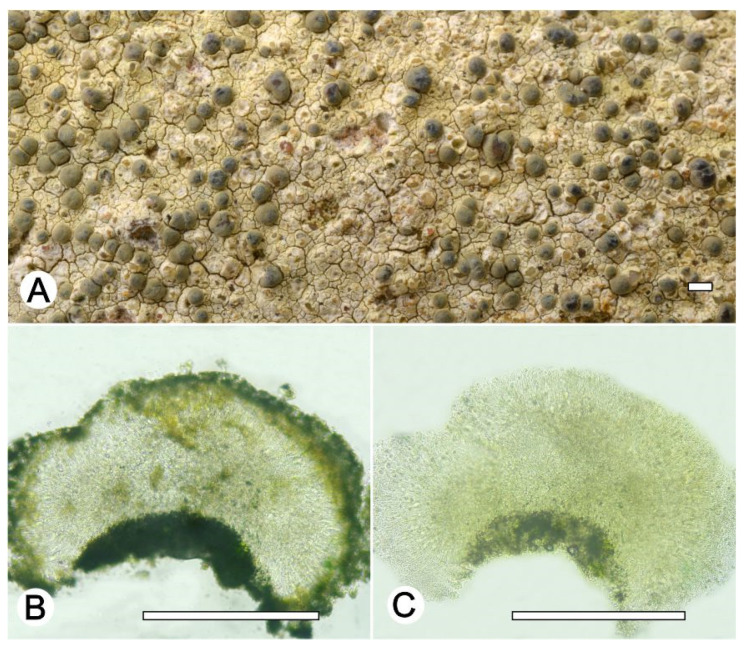
*Lecanora fluorosaxicola* sp. nov. (**A**) Thallus with ascomata (holotype, ISE-54033); (**B**,**C**) Apothecial section after KOH treatment (holotype ISE-54033). Scale bars: (**A**), 1 mm; (**B**,**C**), 100 μm.

**Figure 4 jof-09-00415-f004:**
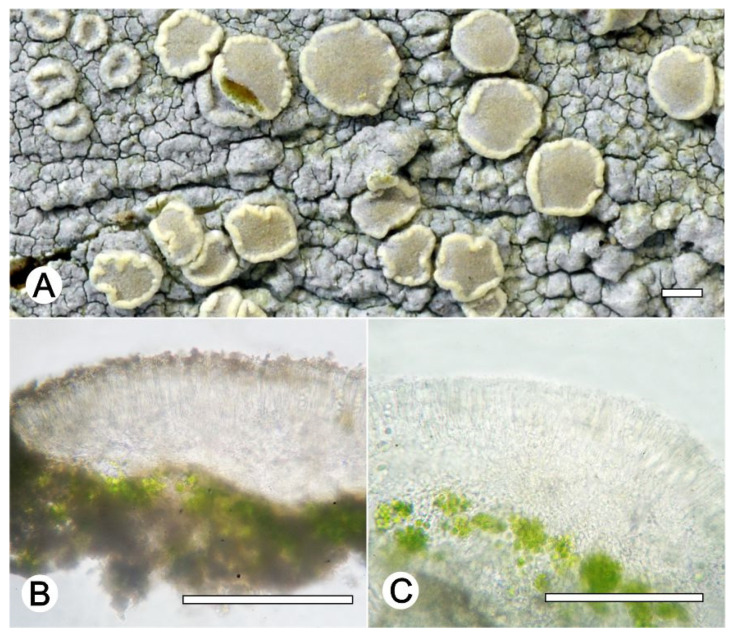
*Lecanora glaucomodes*. (**A**) Thallus with ascomata (ISE-46533); (**B**) Apothecial section in water (ISE-46533); (**C**) Apothecial section after KOH treatment (ISE-46533). Scale bars: (**A**), 1 mm; (**B**,**C**), 100 μm.

**Figure 5 jof-09-00415-f005:**
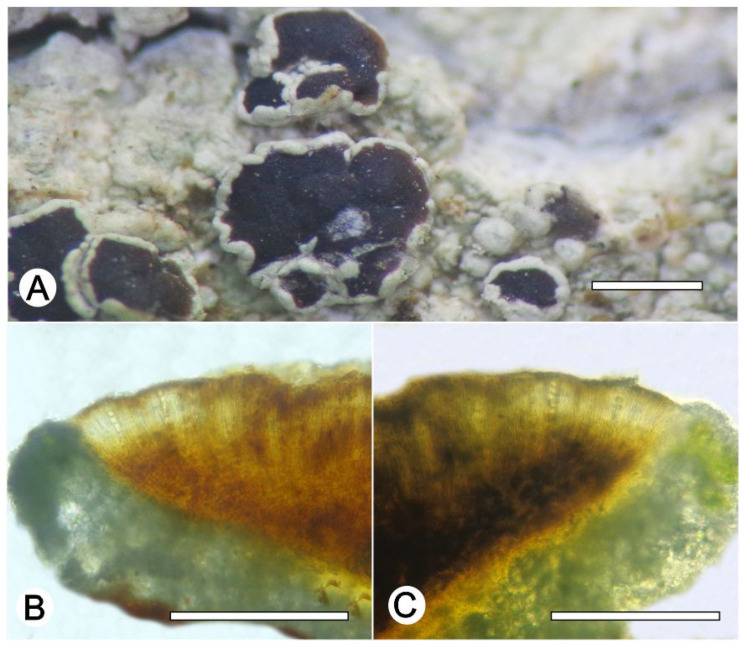
*Lecanora irregularicrocea* sp. nov. (**A**) Thallus with ascomata (holotype ISE-38135); (**B**) Apothecial section in water (holotype ISE-38135); (**C**) Apothecial section after KOH treatment (holotype ISE-38135). Scale bars: (**A**), 1 mm; (**B**,**C**), 100 μm.

**Figure 6 jof-09-00415-f006:**
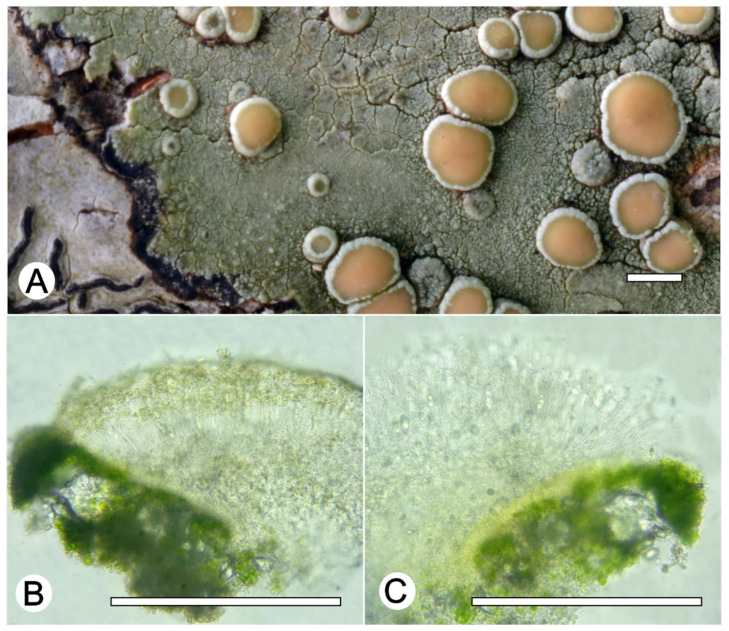
*Lecanora neohelva* sp. nov. (**A**) Thallus with ascomata (holotype ISE-52342); (**B**) Apothecial section in water (holotype ISE-52342); (**C**) Apothecial section after KOH treatment (holotype ISE-52342). Scale bars: (**A**), 1 mm; (**B**,**C**), 100 μm.

**Figure 7 jof-09-00415-f007:**
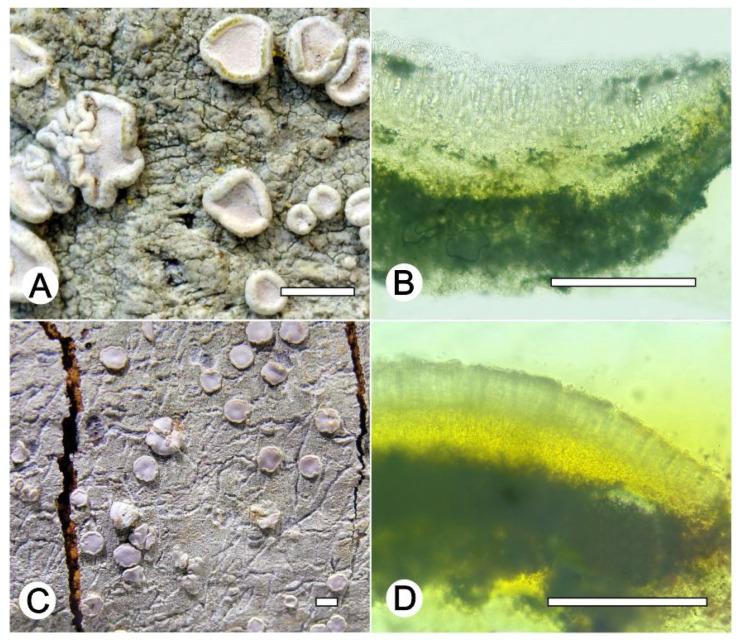
*Lecanora neomerrilli* nom. nov. (**A**) Thallus with ascomata (ISE-38146); (**B**) Apothecial section after KOH treatment (holotype ISE-38146); (**C**) Thallus with ascomata (ISE-52228); (**D**) Apothecial section after KOH treatment (holotype ISE-52228). Scale bars: (**A**–**C**), 1 mm; (**B**–**D**), 100 μm.

**Figure 8 jof-09-00415-f008:**
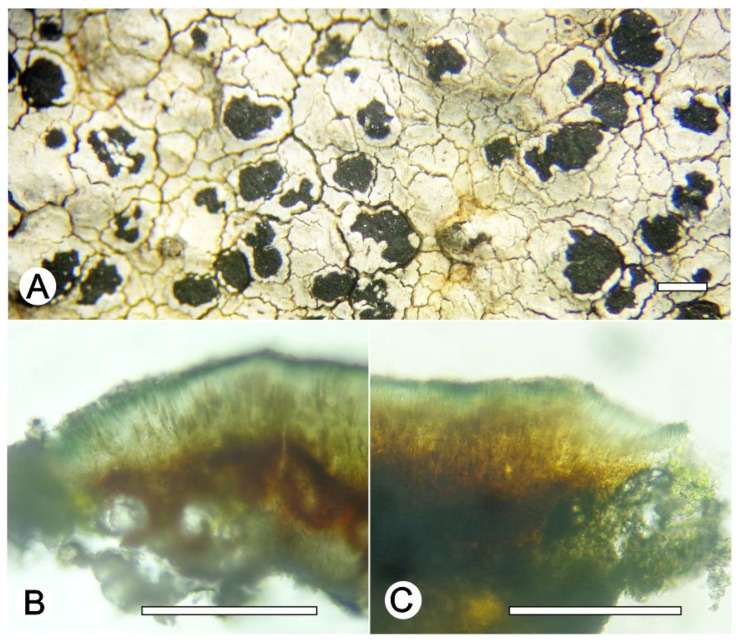
*Lecanora nigrilobulata* sp. nov. (**A**) Thallus with ascomata (holotype ISE-52239); (**B**) Apothecial section in water (holotype ISE-52239); (**C**) Apothecial section after KOH treatment (holotype ISE-52239). Scale bars: (**A**), 1 mm; (**B**,**C**), 100 μm.

**Figure 9 jof-09-00415-f009:**
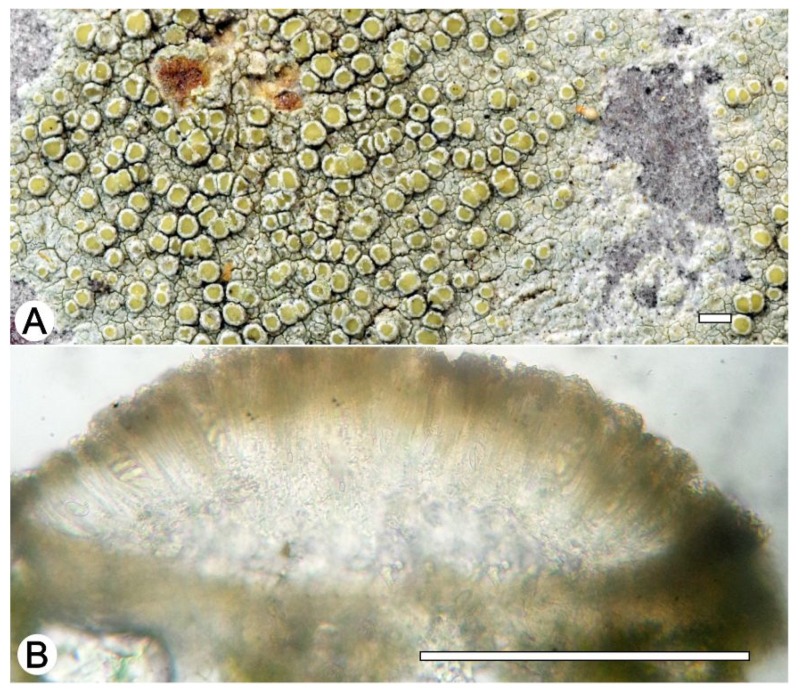
*Lecanora notatictria* sp. nov. (**A**) Thallus with ascomata (holotype ISE-46529); (**B**) Apothecial section in water (holotype ISE-46529). Scale bars: (**A**), 1 mm; (**B**), 100 μm.

**Figure 10 jof-09-00415-f010:**
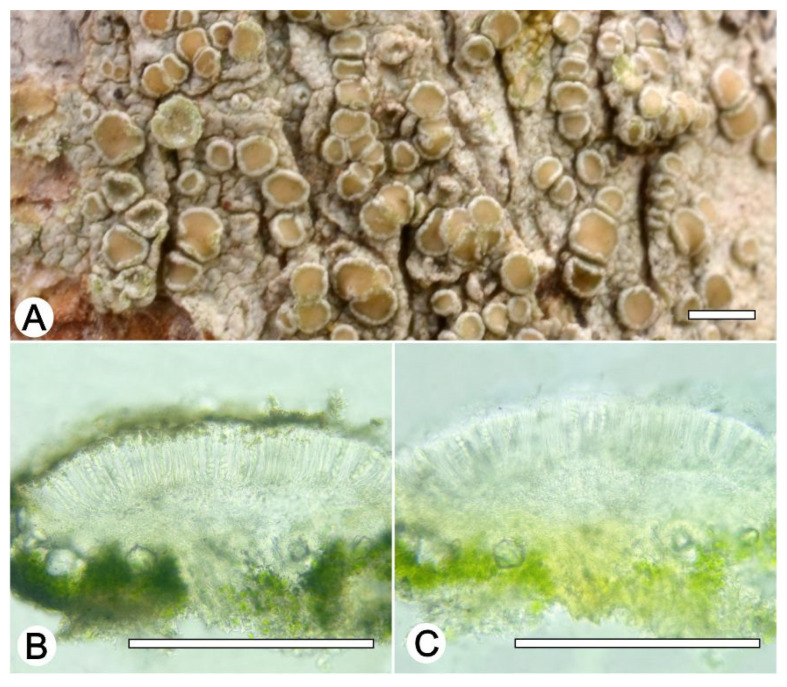
*Lecanora pallidachroa* sp. nov. (**A**) Thallus with ascomata (holotype ISE-54332a); (**B**) Apothecial section in water (holotype ISE-54332a; (**C**) Apothecial section after KOH treatment (holotype ISE-54332a). Scale bars: (**A**), 1 mm; (**B**,**C**), 100 μm.

**Figure 11 jof-09-00415-f011:**
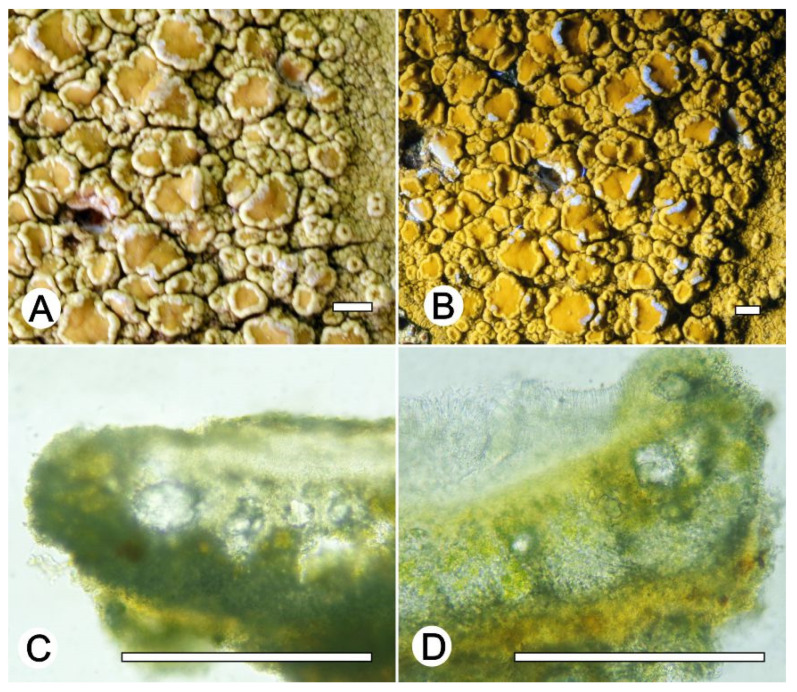
*Lecanora parahelva* sp. nov. (**A**) Thallus with ascomata (holotype ISE-54323); (**B**) Medulla UV+ greenish white (holotype ISE-54323); (**C**) Apothecial section in water (holotype ISE-54323); (**D**) Apothecial section after KOH treatment (holotype ISE-54323). Scale bars: (**A,B**), 1 mm; (**C**,**D**), 100 μm.

**Figure 12 jof-09-00415-f012:**
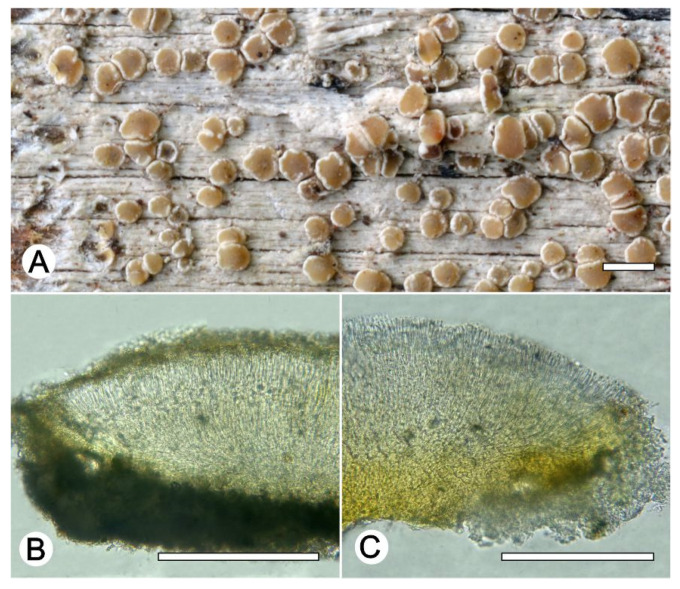
*Lecanora saepiphila* sp. nov. (**A**) Thallus with ascomata (holotype ISE-42612b); (**B**) Apothecial section in water (holotype ISE-42612b); (**C**) Apothecial section after KOH treatment (holotype ISE-42612b). Scale bars: (**A)**, 1 mm; (**B**,**C**), 100 μm.

**Figure 13 jof-09-00415-f013:**
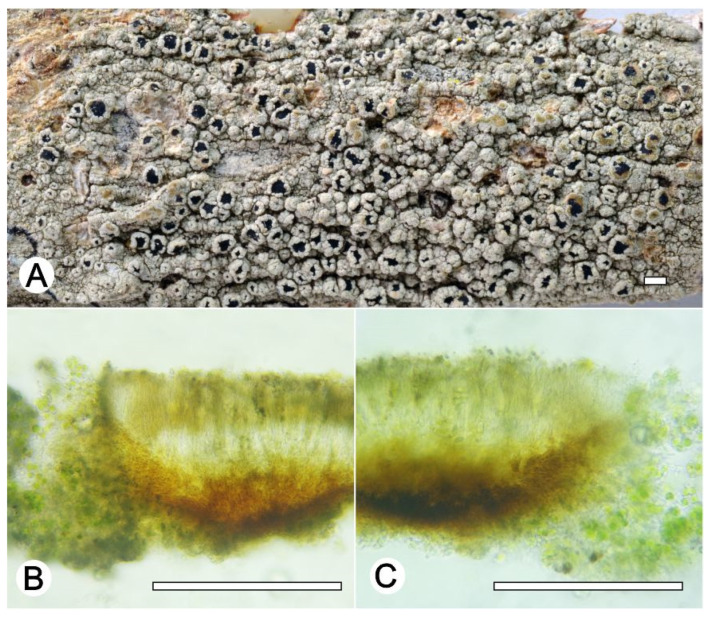
*Lecanora xanthoverrucosa* sp. nov. (**A**) Thallus with ascomata (holotype ISE-52394); (**B**) Apothecial section in water (holotype ISE-52394); (**C**) Apothecial section after KOH treatment (holotype ISE-52394). Scale bars: (**A**), 1 mm; (**B**,**C**), 100 μm.

**Table 1 jof-09-00415-t001:** Gene regions, respective primer pairs and PCR conditions used in the study.

Gene Region	Primers	PCR Condition	References
ITS	ITS1F and ITS4a	95 °C: 5 min, (1) (95 °C: 30 s, 66 °C: 30 s) × 10; (2) (95 °C: 30 s, 56 °C: 30 s, 72 °C: 1 min 30 s) × 34, 72 °C: 10 min	[34,35]
ITS	ITS1F and ITS4	95 °C: 5 min, (95 °C: 30 s, 56 °C: 30 s, 72 °C: 1 min 30 s) × 35, 72 °C for 10 min	[34,36]

## Data Availability

Not applicable.

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
