# Peer review of "Lecanora s.lat. (Ascomycota, Lecanoraceae) in Brazil: DNA Barcoding Coupled with Phenotype Characters Reveals Numerous Novel Species"

_jof, 2023, doi:10.3390/jof9040415_

Round 1

Reviewer 1 Report

Dear authors 

This work is a great contribution of the science

I recommend …

1. The introduction should be expanded, they should mention the state of the art and the most important works from Brazil and the tropical region

2. It is not a requirement, but it is desirable to obtain phylogenies with at least two genes because it is such a diverse genus.

3. Authors should include a discussion on the richness of Lecanora species known from Brazil and indicate in which ecosystem type it is the most diverse and end the manuscript with a conclusion.

4. Provide a taxonomic key of species or a comparative table of characters, since the diagnoses are not very informative

5. The descriptions must be ordered and complete the missing data such as paraphysis, size, shape and color of some structures that are indicated in the first species but it must be extended to all

6. Indicate which author they follow for the terminology used

7. Submit photographs of the spores that, although they are very similar, are always one of the most important characters in the definition of the species.

Author Response

see file

Author Response

all the detailed comments have been followed, including getting mycobank and genbank numbers

Round 2

Reviewer 2 Report

I would appreciate it if the authors explained why the observations were not taken into consideration, how to describe the asci of fungi or include photographs of asci and ascospores.

Author Response

Our descriptions and illustrations mainly consist of characters, especially characters that are diagnostic within the genus. The characters mentioned by the referee, viz. ascus shape and type and ascospore shape, are not variable within the genus, and not even within most of the family. The only character that is variable in this respect is the size of the ascospores, and that is mentioned in each description. There are always 8 ascospores in one ascus and they are always  irregularly biseriately arranged, so measuring the ascus does not add an independant character. In practice, ascospore measurements are much more accurately possible than ascus measurements in any case. The ascus-shape is cylindicoclavate, the type Lecanora-type and the ascospore shape ellipsoid (and hyaline). There are many fungal genera where there are disgnostic characters in these characters, but Lecanora is not one of them.